# *Lactobacillus rhamnosus* TR08 Improves Dyslipidemia in Mice Fed with a High Fat Diet by Regulating the Intestinal Microbiota, Reducing Systemic Inflammatory Response, and Promoting Sphingomholipid Metabolism

**DOI:** 10.3390/molecules27217357

**Published:** 2022-10-29

**Authors:** Xiaohong Feng, Longkun Ding, Guifang Ma, Ying Zhang, Yefu Sun, Zhengzhang Li, Xiaojun Tao, Asmaa Ali, Dongxu Wang, Liang Wu

**Affiliations:** 1Jiangsu Vocational College of Medicine, Yancheng 224005, China; 2Gaoyou Pelple’s Hospital, Yangzhou 225600, China; 3Department of Laboratory Medicine, School of Medicine, Jiangsu University, Zhenjiang 212013, China; 4Department of Pulmonary Medicine, Abbassia Chest Hospital, MOH, Cairo 11517, Egypt; 5School of Grain Science and Technology, Jiangsu University of Science and Technology, Zhenjiang 212100, China

**Keywords:** hyperlipidemia, *L. rhamnosus* TR08 strain, short chain fatty acids, inflammation, metabonomics

## Abstract

Dysbiosis is a crucial manifestation of dyslipidemia; however, oral supplementation of probiotic modulates the intestinal commensal composition. The protective mechanism of probiotics against hyperlipidemia is still under investigation. To elucidate the hypolipidemic effect of *Lactobacillus rhamnosus* TR08 through the analysis of gut microbiota and lipid metabolomics, we investigated changes in gut microbiota and lipid metabolomic phenotypes in mice by real time quantitative PCR and untargeted metabolomics analysis. High fat diet–induced dyslipidemia mice were orally administered with TR08 for 8 weeks. The proinflammatory cytokines (interleukin–2 and interferon–γ) levels in spleen and aortic wall injury in the mice fed with a high-fat diet were inhibited after treatment with TR08 at 1 × 10^8^ CFU per day per mouse. TR08 also reshaped the gut microbiota with increases of the relative abundances of Bifidobacterium and Bacteroides, reduced the abundance of the pro–pathogen bacterial Enterococcus, increased the serum level of short chain fatty acids (SCFAs) contents, and promoted sphingomholipid metabolic pathway. The results indicated that TR08 could improve the intestinal microbiota of mice to increase the production of SCFAs, and then play the anti–inflammation induced by hyperlipidemia and reduce the inflammatory injury of blood vessel wall. Therefore, TR08 can potentially be used as a hypolipidemic effect probiotic in further interventions.

## 1. Introduction

As a consequence of socioeconomic progress in China and changes in lifestyle, the incidence of coronary heart disease (CHD) has increased. The burden of CHD has exceeded that of cancer, and the mortality rate is two times more than cancer-related mortality [1]. One of the important risk factors implicated in CHD is hyperlipidemia. Failure of normal fat metabolism or function due to genetic disease as familial hypercholesterolemia or other diseases such as obesity and diabetes lead to elevating the level of lipids and lipoproteins in the blood [2]. Excess lipids ultimately promote the formation of atherosclerotic plaques in the wall of blood vessels, thus encourage coronary artery stenosis [3], in addition to the direct effect of prolonged hyperlipidemia on myocardium; it inversely endorses the myocardial function and electrophysiological action [4].

The accumulated unhealthy cholesterol underneath the endothelium of blood vessels advocates endothelial dysfunction, which is the first step in atherosclerotic lesion [5]. The latter triggers the endothelial barrier to generate reactive oxygen species (ROS) as well as pro–inflammatory cytokines and chemokines in addition to recruitment of pro–inflammatory leukocyte, thus ending with sub–endothelial transmigration and adhesion [6]. Moreover, oxidative stress in the sub–endothelial layer induces oxidative modification of low-density lipoprotein (LDL) which further results in accumulation of oxLDL in macrophage [7] that eventually leads to excessive apoptosis of macrophage with necrotic core formation; subsequently, the atherosclerotic plaque progresses more and more [8]. On the other hand, several studies point to gathering of excess serum lipids in cardiac muscle, motivating the oxidative stress effect and decreasing the autophagy in addition to significant changes in mitochondrial function of cardiac muscle cells that end with inflammatory cardiac fibrosis and dysfunction [9].

The underlying mechanisms of hyperlipidemia are as yet undiscovered; however, recent studies highlight the role of gut microbiota in the regulation of different body metabolisms in addition to the implication of gut dysbiosis in the development of several chronic diseases, such as chronic kidney disease, obesity, hyperlipidemia, diabetes mellitus, non–alcoholic fatty liver disease, non-alcoholic steatohepatitis, and hypertension [10,11,12,13], and depending on the animal study, the link between gut microbiota and hyperlipidemia and associated disease has been established [14]. Additionally, short chain fatty acids (SCFAs), bile acids (Bas), and lipopolysaccharide (LPS), which are gut microbiota metabolites have been linked to the development of hyperlipidemia [13]. 

Principally, decreasing the level of serum lipids could reverse the sequence of atherosclerosis formation in blood vessels as well the deposition in cardiac muscle cells, thus providing early heart protection [9]. Practically, the hyperlipidemia–lowering agents focus on decreasing total cholesterol (TC) and triglyceride (TG) in blood [13]. However, another newly emerged gut targeting therapy has been proposed in recent studies as prebiotics, probiotics, fecal microbiota transplantation, and herbal medicine as a potential hyperlipidemia–lowering agents [15,16,17,18,19]. *Lactobacillus rhamnosus* strain TR08 (accession No. CCTCC M201654) isolated from human gut was stored in a Microbank at −80 °C. The current study examined the effect of ingestion of probiotics contain *Lactobacillus rhamnosus* TR08 strain on the inflammation of blood vessels wall, expression of pro–inflammatory cytokines in the spleen, and concentration of short chain fatty acid in plasma using a mouse model of hyperlipidemia. 

## 2. Results

### 2.1. L. rhamnosus TR08 Decreased Body Weight and Blood Lipids in Hyperlipidemia Mice

The conventional diet contained crude protein 21.29%, crude fat 21.24%, crude fiber 4.2%; the high fat diet contained 21.13% crude protein, 4.6% crude fat, and 4.4% crude fiber. In the conventional diet, the ratio of protein to energy was 23.28%, the ratio of fat to energy was 11.38%, and the ratio of carbohydrate to energy was 65.34%. The ratio of protein to energy was 19.09%, the ratio of fat to energy was 42.84%, and the ratio of carbohydrate was 38.07% in high fat diet.

The body weights of mice in the HF and HF+LR–TR08 groups were significantly increased at the end of experiment (*p* < 0.05). However, in the NC group, there was insignificant weight gain (*p* > 0.05). Additionally, both the HF and HF+LR–TR08 groups were matched and had a significant difference of body weight compared with the NC group (*p* < 0.05) (Figure 1A). Furthermore, regarding the level of total cholesterol (TC), triglyceride (TG), and low-density lipoprotein-cholesterol (LDL–C), they were significantly higher in both HF and HF+LR–TR08 groups in comparing with the control one (*p* < 0.05); however, high–density lipoprotein–cholesterol (HDL–C) had an insignificant difference between all groups (*p* > 0.05) (Figure 1B).

### 2.2. L. rhamnosus TR08 Attenuated the Inflammation in Blood Vessels Wall of Hyperlipidemia Mice

In the HF group, the vascular lumen of mice aorta was irregular and uneven, the wall was thick, and the cells in the tunica media were disordered without lipid deposition and foam cell aggregation. On the other hand, the features on the HF+LR–TR08 group were quite similar to picture in the NC group (Figure 2).

### 2.3. L. rhamnosus TR08 Influenced the Intestinal Microbiota and SCFAs Production in Hyperlipidemia Mice

The copy numbers of Bifidobacterium and Bacteroides bacteria were significantly increased in the HF+LR–TR08 group compared with both the HF and NC group (*p* < 0.05), while the copy number of *Enterococcus* bacteria was significantly decreased (*p* < 0.05). However, all groups were matched statistically regarding the copy number of *Escherichia* bacteria (*p* > 0.05) (Figure 3a). The three main SCFAs (acetic acid, propionic acid, and butyric acid) were separated by the LC–MS/MS method (Figure 3b). The concentrations of acetic acid, propionic acid, and butyric acid in the plasma of the HF+LR–TR08 group were significantly increased in comparing with both the HF and NC groups (*p* < 0.05) (Figure 3c). 

### 2.4. L. rhamnosus TR08 Reduced the Systemic Inflammatory Response in Hyperlipidemia Mice 

Considering the mRNA expression of pro–inflammatory cytokines interleukin (IL)–2 and interferon (IFN)–γ in spleen tissue; it showed significant downregulation in the HF+LR–TR08 group in comparing with both the HF and NC groups (*p* < 0.05), while anti–inflammatory cytokines IL–4 and IL–10 showed significant upregulation (*p* < 0.05) (Figure 4).

### 2.5. L. rhamnosus TR08 Regulated Metabolomics in Hyperlipidemia Mice

Regarding the untargeted metabolomics analysis, in the HF group, the normal physiological metabolism was significantly disturbed; however, in the HF+LR–TR08 and NC groups, the small fate molecules were easily isolated. Additionally, OPLS–DA analysis showed significant separation of each group (Figure 5A). The cluster heat map and histogram reflected the significant changes of the relative concentration of different metabolites in each group (Table 1 and Figure 5B). In the OPLS–DA model, a series of differentially expressed metabolites were obtained by screening with VIP > 1 and *p* < 0.05. In the HF+LR–TR08 group; the upregulated metabolites were PC (P–15:0/0/0), 4b–hydroxycholesterol, 10–octadecenoic acid and 5–dodecenoic acid, 1 –(2–methoxy–hexadecanyl) –sn–glycero–3–phosphoserine, PA (22:2(13Z,16Z)/16:0), PA (P–20:0/17:2 (9Z,12Z)), PC(P–18:1 (11Z)/ 16:0). However, the downregulated metabolites were 7,10,13,16–docosatetraynoic acid, leukotriene C5, Cer (D18:0/13:0). Furthermore, the sphingolipid metabolic pathway was significantly enriched in the HF+LR–TR08 group (VIP > 0.25) in comparison with HF group (Figure 6).

## 3. Discussion

Diseases such as diabetes mellitus, obesity, atherosclerosis, and chronic heart disease have common features of metabolic disorders, which are chronic inflammatory state [20]. Chronic inflammation is established to be connected with hyperlipidemia [21]. The later is characterized by increasing the blood concentration of TC, TG, and LDL–C, and decreasing the level of HDL–C [22]. The associated risk factors implicated in dyslipidemia are heterogeneous and may be overlapped with each other [10,23]. However, the underlying mechanism of chronic inflammation induced by hyperlipidemia remains unclear. In spite of that, recent studies highlight the link between gut microbiota and hyperlipidemia [24,25,26]. The high fat– and sugar–containing diet affects the diversity of intestinal flora, and could promote the rate of lipid metabolism-related disorders by different mechanisms [10]; this leads to disruption of intestinal integrity that eventually affects the metabolism of cholesterol in liver cells [27].

Moreover, hyperlipidemia–induced dysbiosis can lead to disorder of the local immune system, accompanied by production of inflammatory cytokines such as IL–1β, IL–6, and tumor necrosis factor (TNF)–α, and adipokines such as leptin in addition to increasing the intestinal permeability [28]. Furthermore, it affects the microglia in the brain through “gut–brain axis” leading to neuroinflammation, which in time affects the satiety centers, decreasing the anorexia hormones and producing more craving for food intake and excess body weight [29]. Additionally, the metabolites of gut microbiota act as a signaling pathway in lipid homeostasis [30]. So far, unbalance of microbiota diversity accompanied by excess production of unhealthy metabolites as LPS and trimethylamine N–oxide (TMAO) are closely linked to dyslipidemia [31,32]. Therefore, modification of gut microbiota composition could be considered novel therapeutic approaches in handling the complication of hyperlipidemia.

The present study demonstrated that high fat–containing formulae increased all lipid indices in the blood and induced histopathological changes in blood vessels; the wall became irregular and thick due to distortion of tunica media layers. However, using probiotics containing *L. rhamnosus* TR08 strain provided significant protection against the direct inflammatory effect of hyperlipidemia.

Lactobacillus is a lactic acid bacterium widely found in food and in the intestinal tract, and it has both safe and effective properties [33,34], as it reduces the oxidative stress induced by higher cholesterol, and minimizes the chronic inflammation [35]. In contrast with the current finding, some studies found that some *L. rhamnosus* strains had the ability to lower the level of total blood lipids via different pathways, hence it competes with fatty acids in the intestine and prevents the occurrence of fatty liver [36,37,38], even though, *L. rhamnosus* has different strains with different beneficial modes of action, which are still under investigation.

Our study found that *L. rhamnosus* TR08 increased the number of healthy microbial flora, Bifidobacterium and Bacteroides, which subsequently controlled the chronic inflammatory condition. It encouraged the concentration of SCFAs in serum, the potential microbiota metabolites involved in different anti-inflammatory pathways. It maintained the homeostasis of intestinal mucosa and act as immune modulators, since it activates several immune cascades to minimize the inflammatory process [39]. Lactobacillus, Bifidobacterium, and Bacteroides genus of microbial commensal have the ability to produce SCFAs, which are easily absorbed in colon and reach liver circulation and promote the cholesterol synthesis [40]. However, SCFAs play a role in regulating energy metabolism through the adenosine monophosphate activated protein kinase pathway and the gut–brain axis, and could inhibit the cholesterol and triglycerides synthesis [13].

A recent clinical study found that pregnancy-associated obesity was associated with significant reduction of SCFAs which affect the liver function parameters [41]. Moreover, in patients with Parkinson’s disease, the level of SCFAs was altered, especially propionic acid [42]. Additionally, Duan et al. found that *L. rhamnosus* plays a defensive role in *F. nucleatum*–related colitis through autophagy intervention process [43]. The current data reported that *L. rhamnosus* TR08 reduced the systemic inflammatory response in hyperlipidemia mice. Hence, the mRNA expression of pro–inflammatory cytokines IL–2 and IFN–γ in spleen tissue was significantly downregulated, while mRNA expression of anti-inflammatory cytokines IL–4 and IL–10 was upregulated.

The intestinal cells are directly impacted by probiotics because they are able to modify the expression of several genes that are involved in gut-mediated immunity [44]. However, this action depends mainly on which bacterial strains are used as probiotics. Generally, probiotics provoke a tolerogenic response to external antigens through interaction with Toll–like receptors (TLRs), which subsequently lead to downregulation of mRNA expression of pro–inflammatory cytokines [44]. Another study examined the immune effect of different strains of *L. rhamnosus* and found that they had the ability to induce early pro–inflammatory cytokines as IL–8, IL–6, and TNF–α. Additionally, the phagocytic activity of macrophage was increased in response to treatment with Lactobacillus–containing probiotics [45].

A limitation of the present study is that the sphingomholipid metabolism pathway responsible for the improvement on atherosclerosis of *L. rhamnosus* TR08 was not identified. Therefore, the moderating effect of *L. rhamnosus* TR08 on the sphingomholipid metabolism warrants further research. TLRs as pattern recognition receptor play a key role in pro–inflammatory process, and IL–8, IL–6, and TNF–α are the primary downstream targets of TLRs signaling [44,45]. TLRs inactivation by *L. rhamnosus* TR08 may be attributed to the improvement on atherosclerosis; however, this hypothesis requires further investigation.

This study employs metabolomics analysis for more accurate evaluation of health condition of each treatment group. the data showed that, in the HF group, the normal physiological metabolism was significantly disturbed. On the other hand, in the HF+LR–TR08 and NC groups, the small fate molecules were easily isolated. Moreover, purine, glycerophospholipid, and sphingolipid metabolism pathways were significantly enriched in the HF+LR–TR08 group in comparison with the HF group. This suggests that *L. rhamnosus* is a potential targeted therapy for improving atherosclerosis. The latter was formed due to abnormal lipid metabolic sequence and inflammatory immune response, and manifested by vascular wall dysfunction and plaque formation [46].

## 4. Materials and Methods

### 4.1. High Fat Diet Induced Murine Hyperlipidemia

Thirty 6–week–old healthy male C57BL/6 mice (weighing 18–22 g) were purchased from the Center of Comparative Medicine of Yangzhou University. To analyze the effect of *L. rhamnosus* TR08 strain at preventing and treating the inflammatory lesion of hyperlipidemia, after period of accommodation, the mice were divided into three treatment groups (normal control (NC) group, high–fat diet (HF) group, and high–fat diet + *L. rhamnosus* TR08 strain (HF+LR–TR08) group), with 10 mice in each group. Experimental hyperlipidemia was induced in mice by feeding them high fat diet formula (Table 2) every day; however, the NC group was in standard diet formulae for 8 weeks. In HF+LR–TR08 group, mice were administrated with TR08 (1 × 10^8^ CFU per day per mouse) via gastric tube for 8 weeks simultaneously with induction of experimental hyperlipidemia. The mice in this experiment were fed the conventional diet and the high–fat diet containing 1.2% cholesterol (Product code: XT108C) from Nanjing Synergetic Biology. All mice were fed freely. Additionally, all mice were allowed to drink normal saline freely for 8 weeks. All mice were fed under standard laboratory condition, where the temperature was (25 ± 1) °C, the humidity was (50 ± 10)%, and the light condition was dark/light according to 12 h/12 h cycle.

All animals were subjected to the restricted role of experimental protocol guideline after agreement of Ethics Committee of the Jiangsu University and the rules of animal protection and welfare protocol were obeyed. The body weight of mice in each group was recorded daily, while the plasma, spleen, aortic tissue and colon contents were collected at the end of the experiment.

### 4.2. Histological Analysis

The aortic blood vessel tissue of mice was collected at the end of trial, rinsed with normal saline and fixed with 4% paraformaldehyde. The fixed blood vessels were embedded in wax and dehydrated for cut preparation (6 μm thickness). The section was stained with hematoxylin–eosin (HE) for histological observation to examine the pathology of atherosclerotic inflammatory lesion induced by high fat diet formula.

### 4.3. DNA Extraction and Real Time Polymerase Chain Reaction (qPCR)

To analyze the relative contents of four important intestinal microorganisms of mice colon contents, the samples were collected at the end of experiment and frozen at −80 °C directly. The microbial DNA of mice colon contents was obtained using fecal genomic DNA extraction kit (Beijing Tiangen Biotech CO., Ltd., Beijing, China) according to manufacturer guidelines. The qPCR primers were synthesized by Suzhou GENEWIZ Company, and the primer sequences are shown in (Appendix A). Four bacterial PCR primers were used to amplify the total DNA. The PCR amplification products were ligated with PTG–19 T vector (Shanghai Generay Biotech Co., Ltd., Shanghai, China) and then imported into *E. coli* DH5α strain to be used as plasmid standards. The qPCR standard curves were prepared by gradient dilution of the plasmid standard, and the copy number of plasmids in the standard was 10^2^–10^9^. The qPCR reaction system (Najing Vazyme, China) consisted of 10 μL SYBR Master Mix, 0.4 μL of upstream and downstream primers, and 1 μL of total DNA or plasmid standard (total DNA ≤ 200 ng). The reaction system was supplemented with ddH_2_O to 20 μL. The qPCR reaction conditions were pre–denaturation at 95 °C for 1 min, denaturation at 95 °C for 30 s, and annealing at 55 °C for 30 s. The whole reaction was 40 cycles, and each sample was tested three times. The standard curve was established with Lg copies of standard plasmids of different dilutions as the *x*-axis, and Ct value as the *y*-axis. The copy values of bacteria of four genera were counted according to the Ct values in feces of mice in each group by the standard curve, and the results were expressed as Lg copies/g.

### 4.4. RNA Extraction and qRT–PCR

Total RNA from mice spleen was extracted by standard Trizol method by the total RNA extraction kit (Nanjing Vazyme, Nanjing, China) and reverse transcribed into cDNA by HiScript III RT SuperMix for qPCR (+gDNA wiper) (Nanjing Vazyme, China) in accordance with the instructions, and detected by qPCR. The primers were synthesized by Suzhou GENEWIZ Company as the following sequence (Appendix A). The qPCR reaction system consisted of 10 μL SYBR Master Mix, 0.4 μL of upstream and downstream primers, and 1 μL of cDNA. The reaction was pre–denatured at 95 °C for 20 s, denatured at 95 °C for 5 s, annealed at 60 °C for 20 s, and extended at 72 °C for 30 s for 40 cycles. Each sample was repeated three times, and β–actin gene was used as the internal reference to calculate the relative expressions of each target pro–inflammatory cytokines gene in mouse spleen by formula 2^−ΔΔCt^.

### 4.5. Preparation of Serum Sample

The blood samples of mice were collected at the end of experiment. In order, about 1.5 mL of blood was collected in a tube without any coagulation element, and then was left at room temperature for half an hour before centrifugation at speed 3000× *g*, for 5 min at 4 °C. After that, the serum was wrapped out carefully and transferred to another clean tube to be stored under −80 °C immediately.

### 4.6. Blood Parameters Analysis

The serum levels of TC, TG, low density lipoproteins (LDL–C), and decreasing the level of high–density lipoproteins (HDL–C) were determined using the commercial assay kits from Nanjing Jiancheng Bioengineering Institute (Nanjing, China) according to the manufacturer’s protocols.

### 4.7. Liquid Chromatography–Mass Spectrometry (LC–MS/MS)

The concentration of acetic acid, propionic acid, and butyric acid (the main SCFAs) were measured in mice serum using liquid chromatography–mass spectrometry (LC–MS/MS) in the Test Center of Yangzhou University and verified according to the standard curve of test platform. The brief procedure was as follows: 50 μL of serum was deproteinized with 100 μL of cold–isopropanol and added with 10 μL of 2–isobutoxyacetic acid (2.5 μg/mL). After centrifugation, 100 μL of supernatant was separated in a glass vial for the derivatization step. To the latter were consequently added 50 μL of 50 mM of 3–Nitrophenylhydrazine hydrochloride (3-NPH), 50 μL of 50 mM of 1–Ethyl–3–(3–dimethylaminopropyl) carbodiimide hydrochloride (EDC), and 50 μL of pyridine (7%) in methanol. Derivatization was performed in an incubator at 37 °C for 30 min. The solution was diluted was 250 μL of formic acid (0.5%) in water and directly injected. The LC–MS/MS system consists of an HPLC Dionex 3000 UltiMate system (Thermo Fisher Scientific, Waltham, MA, USA) coupled to a tandem mass spectrometer AB Sciex 3200 QTRAP (Sciex, Milan, Italy) operated under negetive ESI mode. The instrument parameters were: CUR 30, GS1 40, GS2 40, capillary voltage −4.5 kV and source temperature 400 °C. Chromatographic separation was achieved on a Restek Raptor C18 2.7 μm 2.1 × 100 mm (Bellefonte, PA, USA) using as mobile phase (A) water + 0.1% formic acid and (B) acetonitrile. The elution program (%B) was 0–2.5 min 10%, 2.5–16 min 10–50%, 16–16.2 min 50–10% maintained until 18 min. The flow rate was 0.4 mL/min, the column and the autosampler temperature were 35 °C and 15 °C.

### 4.8. Untargeted Metabolomics Analysis

The stored plasma re–melted, hence about 100 μL was absorbed and mixed with 300 μL of methanol. After centrifugation at 16,000× *g* for 10 min, about 120 μL of supernatant was absorbed for detection, in which 10 μL from each sample was mixed and used as a quality control sample (QC). All samples were tested in Yangzhou University center using UPLC–IMS Q–Tof and Unifi software to collect data and match metabolites to molecular formulas, Principal Component Analysis (PCA), and Partial Least Squares Discriminant Analysis (OPLS-DA) were used to analyze the trend of metabolites by Progenesis QI software. The potential differential metabolites were screened according to VIP > 1 and *p* < 0.05 rules. The obtained differential metabolites were retrieved and confirmed in Human Metabolome Database (HMDB). Based on the Kyoto Encyclopedia of Genes and Genomes (KEGG) database, the related metabolic pathways of the potential differential metabolites were determined.

### 4.9. Statistical Analysis

The data were collected in Excel sheet and analyzed using SPSS 22.0 software for windows, the numerical data were expressed as mean and standard deviation (SD), and means differences were compared using one way ANOVA test with post hoc test (Tukey methods). All tests were two-sided, *p* < 0.05 considered significant. For mapping, Graphpad Prism 9.2 software was used.

## 5. Conclusions

In summary, the *L. rhamnosus* TR08 strain may influence the host immune response and protect them against the hazardous of abnormal lipid metabolisms through different ways; it enriched the intestine with good bacterial strains such as Bifidobacterium and Bacteroides, the latter of which augmented the production of effective SCFAs, which act as immunomodulator substances. Moreover, *L. rhamnosus* TR08 strain could regulate the mRNA expression of pro–inflammatory and anti-inflammatory cytokines and thus eventually reduce the chronic inflammation. Another benefit of adding probiotics was to normalize the lipid metabolic pathway by diverting it toward the sphingolipid metabolism track.

## Figures and Tables

**Figure 1 molecules-27-07357-f001:**
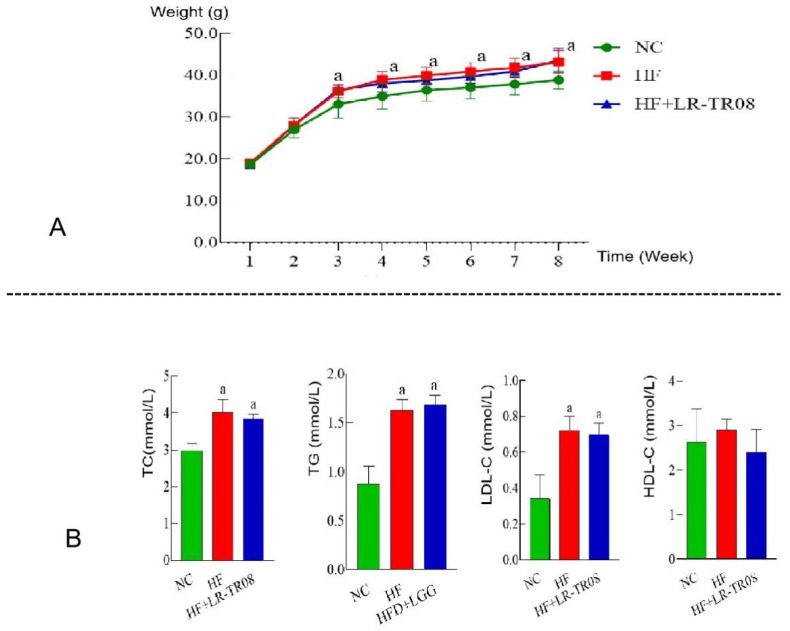
The change of body weight and blood lipid index in HF feed mice. (**A**): the weight change, (**B**): the changes of total cholesterol (TC), triglyceride (TG), low–density lipoprotein–cholesterol (LDL–C), and high–density lipoprotein–cholesterol (HDL–C). a: compared with the NC group, *p* < 0.05. Normal control group: NC, high-fat diet group: HF, and high–fat diet + *L. rhamnosus* TR08 strain group: HF+LR–TR08.

**Figure 2 molecules-27-07357-f002:**
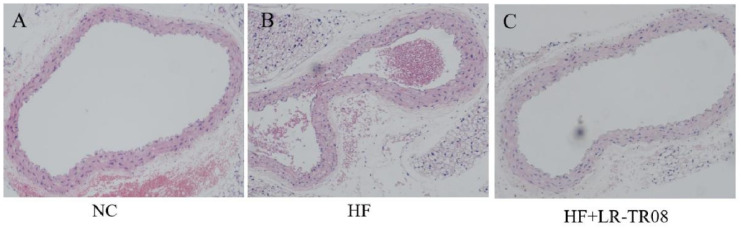
HE staining of mouse aortic tissue. The fixed blood vessels were embedded in wax and dehydrated for cut preparation (6 μm thickness). Normal control group: NC, high–fat diet group: HF, and high–fat diet + *L. rhamnosus* TR08 strain group: HF+LR–TR08.

**Figure 3 molecules-27-07357-f003:**
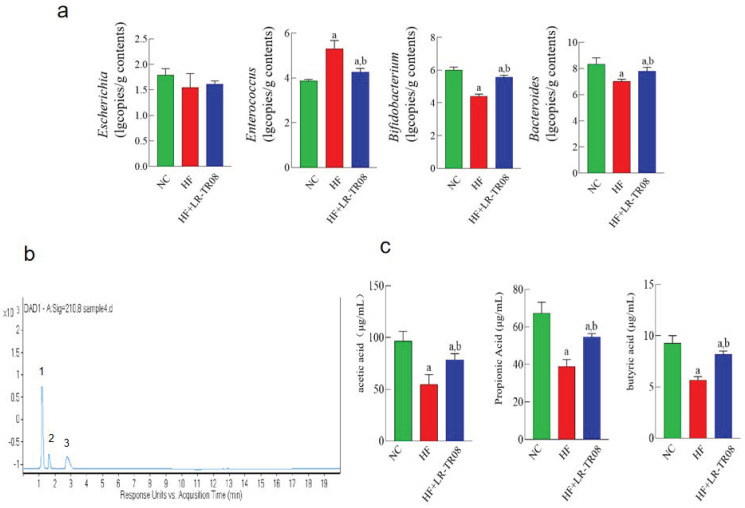
Changes of relative contents of four important bacterial groups (Bifidobacterium, Bacteroides, *Escherichia*, *Enterococcus*) in colon contents of mice and the plasma concentrations of acetic acid, propionic acid and butyric acid. Normal control group: NC, high-fat diet group: HF, and high-fat diet + *L. rhamnosus* TR08 strain group: HF+LR–TR08. (**a**): the relative contents of four important bacteria in colon contents of mice were detected by qPCR (**b**): the ion flow diagram of acetic acid, propionic acid, and butyrate in mouse plasma by LC–MS/MS. (**c**): plasma concentrations of acetic acid, propionic acid, and butyric acid in each group a: compared with the NC group, *p* < 0.05; b: compared with the HF group, *p* < 0.05.

**Figure 4 molecules-27-07357-f004:**
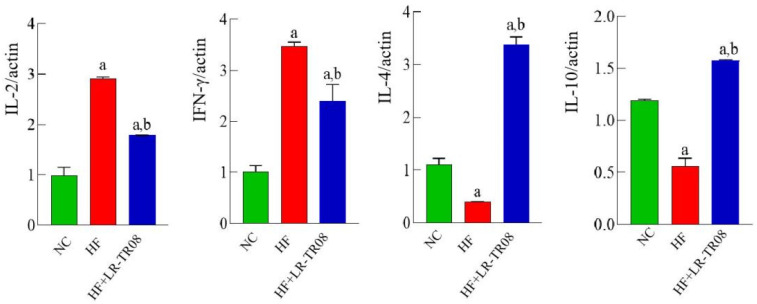
The mRNA expressions of inflammatory factors in mouse spleen. a: compared with the NC group, *p* < 0.05; b: compared with the HF group, *p* < 0.05. Normal control group: NC, high–fat diet group: HF, and high–fat diet + *L. rhamnosus* TR08 strain group: HF+LR–TR08.

**Figure 5 molecules-27-07357-f005:**
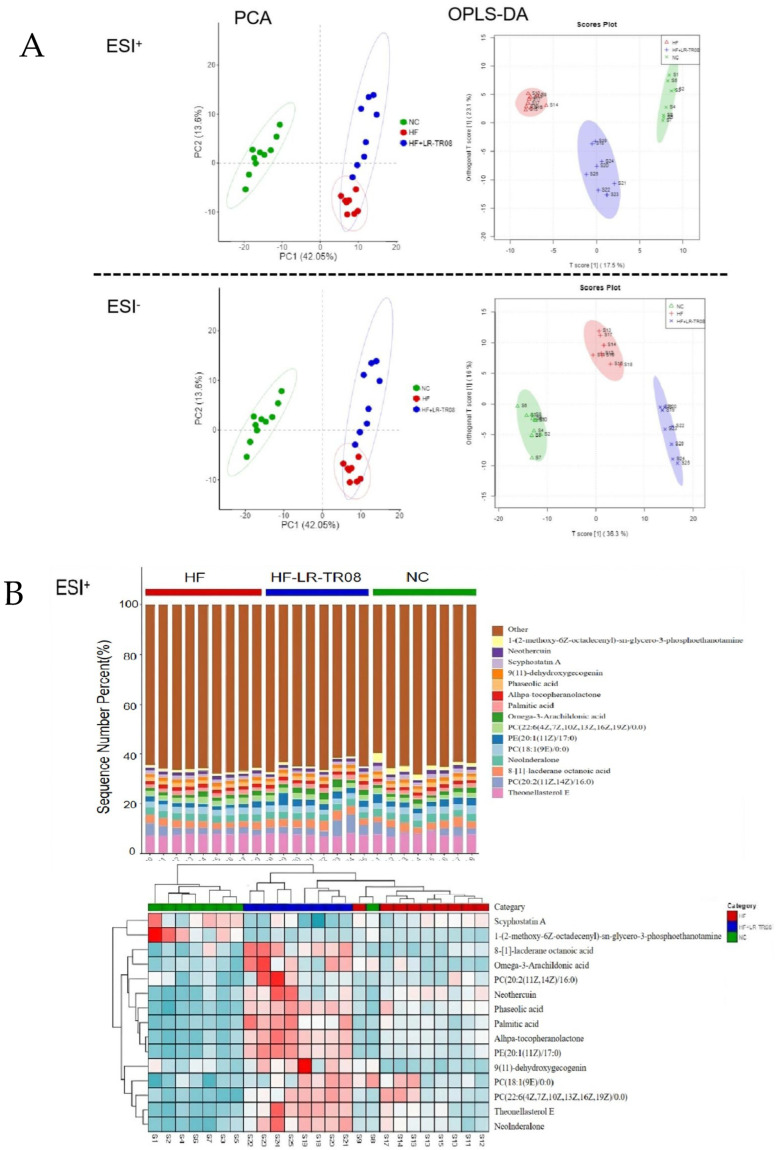
Results of PCA and OPLS–DA (**A**), cluster heat map ESI+ (**B**), and cluster heat map ESI+ (**C**) analysis of metabolites in mouse plasma.

**Figure 6 molecules-27-07357-f006:**
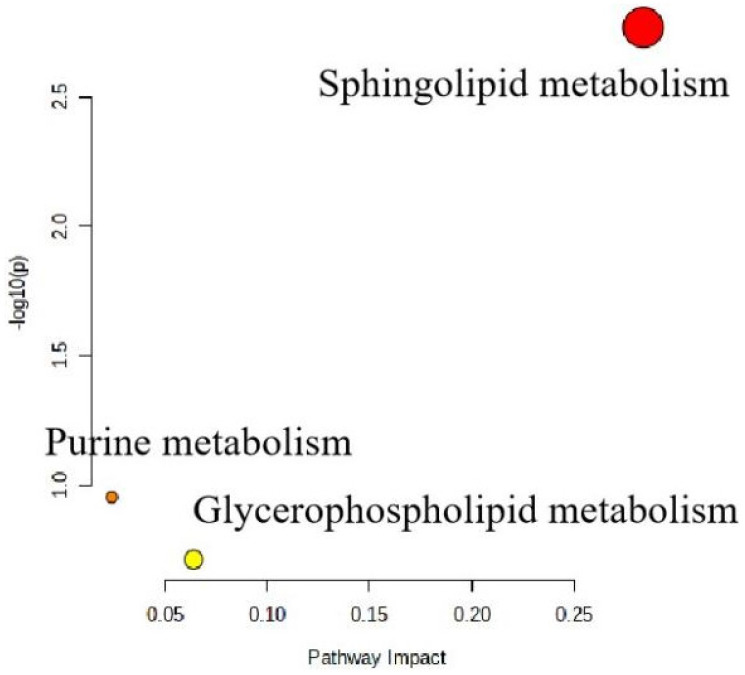
The metabolic pathway analysis of differential metabolites of plasma between the high–fat diet and high–fat diet + *L. rhamnosus* TR08 strain groups.

**Table 1 molecules-27-07357-t001:** The differential metabolites between the high–fat diet and high–fat diet + *L. rhamnosus* TR08 strain groups.

No.	Metabolite	Library ID	Formula	M/Z	RT/min	Trend
1	7,10,13,16–Docosatetraynoic acid	LMFA 01030682	C22H28O2	342.24	27.11	↓
2	PC(P-15:0/0:0)	LMGP0 1070003	C23H48NO6P	466.33	26.64	↑
3	4b–Hydroxycholesterol	HMDB 13643	C27H46O2	447.35	41.88	↑
4	10–Octadecenoic acid	LMFA 01031090	C18H34O2	327.25	36.00	↑
5	5–Dodecenoic acid	HMDB 00529	C12H22O2	593.48	43.07	↑
6	1–(2–methoxy–hexadecanyl)–sn–glycero–3–phosphoserine	LMGP 03060014	C23H48NO9P	534.28	26.37	↑
7	Leukotriene C5	HMDB 12993	C30H45N3O9S	624.29	29.36	↓
8	PA(22:2(13Z,16Z)/16:0)	LMGP 10010761	C41H77O8P	751.52	30.24	↑
9	PA(P–20:0/17:2(9Z,12Z))	LMGP 10030066	C40H75O7P	721.51	32.94	↑
10	PC(P–18:1(11Z)/16:0)	LMGP 01030134	C42H82NO7P	744.59	41.95	↑
11	Cer(d18:0/13:0)	LMSP 02010018	C31H63NO3	520.47	26.81	↓

↑ and ↓: compared with HF group, the trend of metabolite change.

**Table 2 molecules-27-07357-t002:** High–fat diet feed formula.

Ingredients	Content
Cane sugar	20.0%
Lard oil	15.0%
Casein	10.0%
Cholesterol	1.2%
Calcium bicarbonate	0.6%
Land plaster	0.4%
Gunk	0.4%
Sodium cholate	0.2%
Basal feed	52.2%

## Data Availability

The data presented in this study are available on request from the corresponding authors.

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
