# Peer review of "Lactobacillus rhamnosus TR08 Improves Dyslipidemia in Mice Fed with a High Fat Diet by Regulating the Intestinal Microbiota, Reducing Systemic Inflammatory Response, and Promoting Sphingomholipid Metabolism"

_molecules, 2022, doi:10.3390/molecules27217357_

Round 1

Reviewer 1 Report

In this study  the effect of ingestion of probiotics contain Lactobacillus rhamnosus TR08 strain on inflammation of blood vessels wall, expression of pro-inflammatory cytokines in spleen and concentration of short chain fatty acid in plasma using mouse model of hyperlipidemia was examined. 

Both the aim and the selection of methods do not raise any objections, but I have some minors comments:

- a more appropriate term for TC, TG and LDL is "serum / plasma concentration" rather than "level"

- add description for figures

- Figure 5. is not readable

- in section 4. there is no information on diet (the composition of the diet, the preparation method, the fiber content, and the results on intake in each group should be added to Results); how the samples for LC analysis were prepared, how the experiment was finished and under what conditions the biological material was collected

- lines 356 and 360 -  The Declaration of Helsinki is about humans, not animal research.

Moreover the discussion must be improved. Authors should indicate their main results and conduct a structured discussion about the results of their own research. There is also no indication of strengths and limitations

Author Response

Reviewer 1#

In this study the effect of ingestion of probiotics contain Lactobacillus rhamnosus TR08 strain on inflammation of blood vessels wall, expression of pro-inflammatory cytokines in spleen and concentration of short chain fatty acid in plasma using mouse model of hyperlipidemia was examined. Both the aim and the selection of methods do not raise any objections, but I have some minors comments:

- a more appropriate term for TC, TG and LDL is "serum / plasma concentration" rather than "level"

Response: Revised. Thanks.

- add description for figures

Response: Thanks for your comments. Revised.

- Figure 5. is not readable

Response: We replaced the image with a higher resolution. Thanks.

- in section 4. there is no information on diet (the composition of the diet, the preparation method, the fiber content, and the results on intake in each group should be added to Results); how the samples for LC analysis were prepared, how the experiment was finished and under what conditions the biological material was collected

Response:  The mice in this experiment were fed the conventional diet and the high-fat diet containing 1.2% cholesterol (Product code: XT108C) from Nanjing Synergetic Biology. We fed the mice freely. The feed was prepared in a way the company did not agree to provide, citing trade secrets. The conventional diet contained crude protein 21.29%, crude fat 21.24%, crude fiber 4.2%; the high fat diet contained 21.13% crude protein, 4.6% crude fat and 4.4% crude fiber. In the conventional diet, the ratio of protein to energy was 23.28%, the ratio of fat to energy was 11.38% and the ratio of carbohydrate to energy was 65.34%. The ratio of protein to energy was 19.09%, the ratio of fat to energy was 42.84% and the ratio of carbohydrate was 38.07% in high fat diet. In our this study, non-targeted metabolomics and the detection of three major SCFAs were performed by the Testing Center of Yangzhou University. The SCFAs were detected by LC-MS/MS method. For the serum samples used for testing. We strictly fixed the serum precipitation time to prevent the loss of volatile substances from excessive blood. The isolated serum was stored at -80℃ and prevented from repeated freezing and thawing.

- lines 356 and 360 - The Declaration of Helsinki is about humans, not animal research.

Response: We apologize for this mistake, revised. Thanks.

Moreover the discussion must be improved. Authors should indicate their main results and conduct a structured discussion about the results of their own research. There is also no indication of strengths and limitations.

Response: We thank the reviewer for pointing out this issue. We have revised the relevant content in the Discussion.

Reviewer 2 Report

This paper has interesting and important information about the influence of Lactobacillus rhamnosus TR08 on dyslipidemia in mice fed with a high fat diet. The results showed that TR08 could improve the intestinal microbiota of mice to increase the production of SCFAs, and then play the antiinflammation induced by hyperlipidemia and reduce the inflammatory injury of blood vessel wall.

However, there are issues that need to be addressed:

1. Name chronic diseases. Which? (line 65)

2. line 67 - Specify which disease it is about

3. Figure 1-4 - No explanation of abbreviations - should be completed.

4. Figure 2 - No group explanations.

4. Section 4.7. No information about the method of preparing the analysis for tests and the determination procedure was not described - it should be completed. What were the operating conditions of the chromatograph? This information is important and should be described.

5. Table 1 - No explanation of abbreviations.

Author Response

This paper has interesting and important information about the influence of Lactobacillus rhamnosus TR08 on dyslipidemia in mice fed with a high fat diet. The results showed that TR08 could improve the intestinal microbiota of mice to increase the production of SCFAs, and then play the antiinflammation induced by hyperlipidemia and reduce the inflammatory injury of blood vessel wall.

However, there are issues that need to be addressed:

  1. Name chronic diseases. Which? (line 65)

Response: Thanks for your comment. Revised.

  1. line 67 - Specify which disease it is about

Response: Thanks for your comment. Revised.

  1. Figure 1-4 - No explanation of abbreviations - should be completed.

Response: Thanks for your comment. Revised.

  1. Figure 2 - No group explanations.

Response: Thanks for your comment. Revised.

  1. Section 4.7. No information about the method of preparing the analysis for tests and the determination procedure was not described - it should be completed. What were the operating conditions of the chromatograph? This information is important and should be described.

Response: Added. Thanks.

  1. Table 1 - No explanation of abbreviations.

Response: Thanks for your comment. Revised.

Round 2

Reviewer 2 Report

I accept in present form.